# Phase-Change Materials in Hydronic Heating and Cooling Systems: A Literature Review

**DOI:** 10.3390/ma13132971

**Published:** 2020-07-03

**Authors:** Rok Koželj, Eneja Osterman, Fabrizio Leonforte, Claudio Del Pero, Alessandro Miglioli, Eva Zavrl, Rok Stropnik, Niccolò Aste, Uroš Stritih

**Affiliations:** 1Faculty of Mechanical Engineering, University of Ljubljana, Aškerčeva 6, 1000 Ljubljana, Slovenia; eneja.osterman@fs.uni-lj.si (E.O.); eva.zavrl@fs.uni-lj.si (E.Z.); rok.stropnik@fs.uni-lj.si (R.S.); uros.stritih@fs.uni-lj.si (U.S.); 2Department of Architecture, Built Environment and Construction Engineering, Politecnico di Milano, Via Edoardo Bonardi, 9, 20133 Milano, Italy; fabrizio.leonforte@polimi.it (F.L.); claudio.delpero@polimi.it (C.D.P.); alessandro.miglioli@polimi.it (A.M.); niccolo.aste@polimi.it (N.A.)

**Keywords:** PCM thermal storage, latent thermal storage, latent heat storage, PCMs in hydronic systems, PCMs for heating, PCMs for cooling, PCM heat storage, PCM cold storage, PCM maturity, TRL of PCM

## Abstract

When considering the deployment of renewable energy sources in systems, the challenge of their utilization comes from their time instability when a mismatch between production and demand occurs. With the integration of thermal storages into systems that utilize renewable energy sources, such mismatch can be evened out. The use of phase-change materials (PCMs) as thermal storage has a theoretical advantage over the sensible one because of their high latent heat that is released or accumulated during the phase-change process. Therefore, the present paper is a review of latent thermal storages in hydronic systems for heating, cooling and domestic hot water in buildings. The work aims to offer an overview on applications of latent thermal storages coupled with heat pumps and solar collectors. The review shows that phase-change materials improve the release of heat from thermal storage and can supply heat or cold at a desired temperature level for longer time periods. The PCM review ends with the results from one of the Horizon2020 research projects, where indirect electrical storage in the form of thermal storage is considered. The review is a technological outline of the current state-of-the-art technology that could serve as a knowledge base for the practical implementation of latent thermal storages. The paper ends with an overview of energy storage maturity and the objectives from different roadmaps of European bodies.

## 1. Introduction—The Use of Energy Storage in Buildings

Buildings’ CO_2_ operational emissions account for 30% of the total energy-related carbon emissions [1]. Indeed, the decarbonization of the building sector plays a central role in the action against climate change. The reduction in building carbon emissions goes through the improvement of energy efficiency, the electrification of the building final energy consumptions and the spread of renewable energy sources (RES) [2]. A higher RES penetration and the increase in building-related electricity consumption require a smarter energy management at the building level, as outlined by the revised Energy Performance of Buildings Directive (EPBD 2018/844/EU) [3]. In fact, the integration of RES introduces several problems in the management of electric systems, since renewables that are more easily integrable in buildings have unpredictable energy production profiles and high variable rates (e.g., solar energy) [4]. Thus, as RES integration increases in the building sector, the need to properly manage and dispatch energy at the building/district level becomes crucial [5]: buildings must be able to balance their on-site energy generation and consumption. In such a context, the use of electricity-driven heat pump systems for heating, ventilation and air-conditioning (HVAC) needs, combined with photovoltaic (PV), is one of the major options to increase RES coverage of building primary energy consumptions [6]. However, the increase in electricity consumptions and the extensive diffusion of non-programmable RES technologies may overload the power grid, thus requiring strategies for a reduction in peak demand [7] and demand response [8]. The reason is attributable to the different distribution of thermal loads and solar radiation on both a daily and seasonal base, making it necessary to exchange electricity with the grid [9], with the consequent reduction in local RES self-consumption. An energy storage system (ESS) is a solution to store the excess of energy, to be exploited at a second time of the day or even later, reducing the mismatch between renewable energy production and the building’s energy consumption [10], thus ensuring energy flexibility on the demand-side [11,12].

Evidence of the increasing interest in such a topic is to be found in the constant growth of the scientific literature [13], in the projects funded at the European level [14,15] and in the new H2020 calls [16], together with the implementation of specific tasks by the International Energy Agency [17]. ESS deployment in 2018 has almost doubled compared with the previous year, with “Behind-the-meter“ almost tripled with respect to 2017 [18], in line with the growth needed to reach the Sustainable Development Scenario (SDS) goals in 2030 [19]. The market for ESSs is indeed expected to grow 9-fold from 2017 to 2022 [20], driven by an almost 75% decrease in ESS prices over the last five years [21]; however, the reason of such success of ESSs is not only economic. ESSs, in fact, are an effective solution for load matching in buildings, as well as for the flexible management of energy fluxes. The presence of storage allows to increase the self-consumption of local RES production and to reduce the exchange with the grid—and thus the related costs—with both energy and financial benefits [22]. Different energy storage technologies are available for power storage, namely mechanical, electrical, chemical and thermal storages [23]. In general, it is possible to state that the most common ESSs in the building sector are electrochemical [24]. Even if thermal storage (TS) is a cheaper and very promising solution to store PV energy in the form of heat, if coupled with heat pump (HP) systems, it is indeed possible to convert the excess of PV electricity in energy in the form of chilled or heated water, which can be exploited for heating or cooling purposes, depending on the season [25]. In such respect, both sensible thermal storage (STS) and latent thermal storage (LTS) can effectively reduce the daily mismatch between solar or wind electricity production and building thermal loads. Of course, according to the specific application context, the adoption of long-term instead of short-term storage solutions allows to address the seasonal mismatch also, with advantages from both energy and economic perspectives [26]. However, due to their lower complexity and cost, short-term storages could be considered the best technologies available for building applications.

In such a context, this paper focuses on short-term TS applications with phase-change materials (PCMs) in hydronic systems for heating/cooling/domestic hot water (DHW) in buildings, with a focus on heat pumps (HPs) and solar technology as heat or cold generators. The use of phase-change materials (PCMs) as TS, also called latent thermal storage (LTS), has a theoretical advantage over sensible thermal storage (STS) because of its high latent heat that is released or accumulated during the phase-change process, which takes place at a nearly isothermal temperature [27]. PCMs have gained popularity for building applications in terms of energy storage density [28], and also flexibility for thermal storage and supply, when coupled with HPs [29]. Therefore, the present paper attempts to review the state-of-the-art technologies, with a focus on HPs and solar technology for LTS applications in buildings’ hydronic systems for heating/cooling/DHW. A review of a TS system is presented in Section 2, representing TS technology with PCMs as storage materials and water as a heat transfer fluid (HTF). This section ends with a review of a novel heating and cooling system, with an indirect electrical storage in LTS through HP, developed within a Horizon 2020 project. Section 3 ends with a review of the LTS maturity on the market and the future objectives for LTS.

## 2. Review of Thermal Storages with PCM in Building’s Hydronic Systems

As already mentioned, the present article focuses on LTS associated with hydronic systems for space cooling and heating in buildings. The size of such storage units can vary quite a lot for larger buildings, whereas for smaller ones, they are typically in the range of up to 100 kWh [30]. The amount of energy stored depends on the size of the storage tank and the type of PCM used, while the heat power during discharge depends on the design of the storage tank. Typical operating temperatures of such systems are between 0 and 15 °C for cooling, and between 20 and 100 °C for heating [30].

The article deals with short-term storage, which helps to install HVAC devices with smaller cooling/heating powers and to reduce the use of energy. In addition to size, temperature level and time, also energy density plays an important role, due to spatial constraints, which are increasingly becoming a problem. A recent work [24] identified the key performance indicators of energy storage systems in order to simplify the comparison of such systems. Key performance indicators (KPIs) were applied to 10 case studies (not only latent heat storage, but also sensible and thermochemical), and one of the findings was that some indicators could not be determined with sufficient reliability, since the information requested for their calculation was not available. The same observation holds true for this article, since not all data are available in the referenced articles.

In this section, the review of TS applications including PCMs (LTS applications) in hydronic systems for heating and cooling of buildings is divided into three sub-sections, each representing a different type of the following system operation (Figure 1): 2.1, space cooling; 2.2, space heating; and 2.3, space heating and cooling. The data obtained from a review of LTS applications in hydronic systems for each particular system operation are summarized in each sub-section in Table 1 (Section 2.1), Table 2 (Section 2.2) and Table 3 (Section 2.3), according to: technology type (e.g., water/water HP, solar collector, etc.), encapsulation (e.g., cylindrical, plates, etc.) and material type (e.g., salt hydrate, paraffin, etc.), melting point of PCM, material and thermal properties of PCM in liquid (first value) and solid (second value) state such as latent heat of fusion (*h*), specific heat of PCM (*c_p_*), thermal conductivity of PCM (*λ*) and density of PCM (*ρ*). Furthermore, the overview in the tables continues with data of accumulated heat/cold, size of storage system, energy density and coefficient of performance (COP) or seasonal performance factor (SPF). This kind of summarized overview provides a comparison between different LTS applications, with a focus on improved system performance, and consequently reduced energy use in hydronic systems for heating and cooling of buildings. A normalized comparison between different sizes of TS systems was made, by providing energy density according to the TS utilized in the applications considered. In the last section, Section 2.4, a conceptual design of TS for heating and cooling in the HEART system is reviewed, and an application with the inclusion of PCMs for indirect electrical storage in the form of heat or cold is presented.

### 2.1. Space Cooling

Chiller plants are mostly used in commercial buildings to generate cold water for air conditioning. They tend to be oversized in terms of maximum design load, therefore they definitely lead to higher energy consumption compared with a properly sized cooling system [31]. LTS can be installed to avoid these problems; in addition, they produce cold in off-peak periods, and thus use off-peak electricity fares.

Several studies have already analyzed the use of PCMs as a storage medium in cooling systems. An extensive review of PCMs in terms of material properties, encapsulation and heat transfer enhancement was carried out by Li et al. [32,33]. There is a number of studies focusing on ice [34,35,36] and the outcomes are positive, meaning that the inclusion of ice brings energy savings [37,38]. However, PCMs with higher melting temperatures are more suitable to work with standard air-conditioning systems because they are able to work with heat transfer fluid (HTF) temperatures above 0 °C. Thus, according to [39], PCMs with a melting/solidification temperature around 8 °C are suitable materials for these applications. In the following chapter, systems with HP in connection with PCM are analyzed, whereas solar energy systems are presented separately, since they usually provide both heat and cold. There are also other technologies, such as direct evaporative cooling [40] or dry coolers [41,42], but their review will be omitted because here the focus is on HP and solar energy.

#### Systems with HP

Reversible HP can be used both for the cooling and heating of buildings: they are efficient systems, generally electrically driven using air, ground or water as a thermal source. To improve the COP of HP, they are coupled with LTS, which consequently leads to a reduction in size and costs [43,44].

In this sub-section, different approaches to the installation of a HP and PCM storage tanks into a hydronic system are presented. With such an arrangement, one can take advantage of the low ambient temperatures during the night to cool a storage tank with a high COP [43,45]. Such a kind of tank can be used later to cool the building, when the outside temperature rises. Another option is the use of cold TS for shifting the cooling load. One such example was experimentally tested by Moreno et al. [46]. Water/water HPs coupled with TS tanks were experimentally investigated for shifting the cooling load of a small house-like structure and its influence on air supply for maintaining indoor temperature within comfort levels. They compared two different configurations: a water tank and a PCM tank. The latter was stacked with plates filled with salt hydrate (S10), with a melting temperature of 10 °C. The water storage tank had a volume of 104 L and the PCM storage tank had a volume of 104 L, with 56 L of water (works as TS and HTF) and with 48 L of PCM, which represents 46% of the volume. Two tests of each tank were performed, and the results showed that the LTS tank (water with PCM) had stored a 25% to 46% higher amount of cold (35.5% on average). However, because of the discharge efficiency, which is 18% higher in the water TS case, the supply of cold in the LTS case was only 11% to 18% higher (14.5% on average) than in the water TS case. Further, the indoor temperature could be maintained in the LTS case for 21% longer than in the water TS case.

A quite similar system was installed on the Solar Decathlon Europe 2012 competition [47]. Their system included a cold and hot storage tank with flat containers filled with salt hydrate, as PCM. The objective was to improve the performance of a water/water HP with the integration of the storage tanks considered. Therefore, the authors investigated the influence of the shifted cooling load with cold storage (10 °C) and the influence of the use of a hot storage tank (27 °C) as a HP’s heat sink at a lower temperature, in order to improve the COP of HP without storages. The results showed an approximately 19% saving in electricity use for the HP’s operation. The improvement of the HP’s COP was studied, but the authors presented the COP results only in graphs with and without a warm tank. From the results, one can graphically see that the COP improved with the use of a warm tank, although the values were not stated.

The next system to consider is a HVAC, whose novelty is a single hydronic circuit for the simultaneous heating and cooling of buildings with water temperatures of about 22 °C all year round [48]. The authors analyzed four different configurations, taking a district heating/cooling network as a baseline, while the rest of the configuration was a reversible air-to-water HP coupled with free cooling devices. They developed a model of a 21 m^3^ PCM-based heat exchanger as cold storage in Modelica. The PCM was plant-based, and had a melting temperature of 18 °C. They simulated the impact of storing cold during the night and supplying it during the day. The results showed that such integration reduces the annual primary energy of about 67% in comparison with a baseline thermal plant. In addition, the use of the PCM-based heat exchanger allowed to avoid the need of mechanical cooling for almost the entire year, by storing 1.1 MWh of cold, which was sufficient to meet the entire cooling load of 806 kWh during the daytime. Only on a few days in the summer, where the lowest amount of stored cold was 249 kWh, the activation of the HP was required.

Another interesting system is “geothermal free cooling”, also known as “geocooling”, in combination with TS [49]. The authors developed and validated a spherically encapsulated PCM tank model and carried out simulations within TRNSYS for a lightweight commercial building located in a Mediterranean climate. The base case was a ground-source heat pump (GSHP), while other cases were direct geocooling with/without a PCM tank. The PCM in spherical capsules was a mixture of water and nucleating agents with a melting point at 0 °C, and the size of the LTS tank was 500 L, with 300 L of PCM, representing 60% of the volume. Simulations showed that geocooling alone meets the cooling loads for 84% of the occupied hours in a four-month cooling season. Therefore, supplementary cooling was required for 180 h out of a total of 1098 occupied hours (from 8 a.m. to 6 p.m.), spread over 40 separate days (thus 16% of the time period considered). The addition of LTS in geocooling can meet the cooling loads for over 99% of the time in the same period. Using the crudest LTS control strategy, which is realistic for practical use, electricity use savings over GSHP are 24% to 45%. The system performance factors (SPF) of the base system, geocooling and geocooling with LTS are 3.0, 5.0 and 5.1, respectively. Therefore, the improvement is 41% higher than the base case, and only 2% higher than geocooling without TS. The latter difference is small, due to the additional energy requirement for charging and discharging the LTS. Despite obtaining results of electricity savings for geocooling with LTS over GSHP, the research did not provide the results for geocooling without TS, which would show more credible comparison between savings.

There are several more studies investigating the energy-saving performance of an active PCM system in buildings using prototype-scale experiments and numerical assessments, and they more or less only induce positive effects on the systems (sometimes without a critical view on results). However, performance in the operational phase of a real building is less understood, so Alam et al. [50] assessed the energy-saving performance of a PCM system installed in an eleven-story building in Melbourne, in order to minimize the daytime cooling load on the electric chiller. This was done with cold storage in LTS from ambient air temperature during the night, using an adiabatic cooler. A hydrated salt PCM with a melting temperature point at 15 °C was encapsulated in 5120 flat panels and stacked in 40 m^3^. The PCM represented a 20 m^3^ volume in the storage tank (50%). They monitored temperatures for 25 months, and found out the following: the PCM system reduced the cooling energy demand on the chiller by 12–37% only during winter months. In summer, the system remained inactive because the ambient temperature was too high to charge the PCM tank during the night. It reached only 15% of its total capacity during summer, and there were occasions when more energy was consumed by the pumps than that which was actually stored in the PCM tank. Considering the latent capacity (stated by manufacturer) of 1307 kWh, it could be stated that the LTS reached its full potential during some of the winter months, where the maximum cold discharged from the LTS was approximately 1500 kWh. Considering that the cooling load of the building during that month was approximately 42,000 kWh, it follows that the full potential of LTS can only cover 37% of the cooling load during winter. This also means that during the maximum cooling load in summer, a fully charged LTS will cover only 1% of the cooling load. This result clearly indicates the need to thoroughly consider the size of the storage and the configuration of the control strategy. In this case, the adiabatic cooling and melting temperature of 15 °C represented a mismatch in this system, because PCM were not able to solidify during the summer nights as the ambient air temperature was higher than the phase-change temperature.

An overview of the described cooling technologies with integrated PCM applications in Section 2.1 is presented in Table 1. The inclusion of PCM positively affects the system in all examples; however, the baseline systems to which LTS applications were compared do not always necessarily represent the best possible comparison. This means that a comparison with the inclusion of PCMs was made with systems without TS (second, third and fourth example) instead of comparing them to conventional TS, like water TS units, which would provide a more reliable comparison. In addition, the LTS in the fifth example was not correctly designed. Therefore, it does not represent the actual potential that the LTS could have in a real environment, but it only emphasizes the importance of a correct approach to designing such an application. Nevertheless, in the first example, the comparison was made between water storage tanks with and without PCMs. It was found out that 46% of PCM encapsulated in panels and implemented in the water storage tank increased the cold supply by up to 18%, and by 14.5% on average. In addition, this application had one of the lowest energy densities (25 kWh/m^3^) and the lowest percentage of PCM (46%) in a TS system among the reviewed technologies of this section. It could be concluded that the reason for the best performance lies in a higher efficiency of discharge, which could be the consequence of a greater heat transfer surface between PCM and water, considering the smaller amount of PCM.

### 2.2. Space Heating

Similarly to the classification of cooling systems, the same can be done to heating systems. In this case, the expected melting temperatures of PCMs are higher.

#### 2.2.1. System with HP

In order to increase the heat storage capacity and improve the performance of the system, HPs are being equipped with TS systems. In this section, different implementations of a PCM storage tank connected to a HP will be presented. The first example is an air-source HP water heater (HPWH), either with a water tank (138 L) or with LTS (149 L). Zou et al. [51] set up an experimental rig where 11 L of PCM (7.5% of volume) was filled in the space between the outer wall of the water tank and its casing. A condenser coil was wrapped around the outer wall of the water tank, where also fins were welded, to increase the heat transfer of the surrounding PCM. The PCM was paraffin with a melting temperature at 43 °C and the operating temperature of TS had a temperature difference of 40 K. An analysis revealed that this implementation has several positive effects on the system: 14% increased heat release, 5% higher COP of HPWH (3.74), 12% electricity savings and 13% shortened charging time. These results are the average of the HPWH with PCM for two cycle operations, where the performance of HPWH with a standard water tank was better than that of the first operation with PCM. The reason is that the heat stored by PCM at the first operation is discharged into water at the second operation, which then reduces the electric power needed for the second operation cycle. Therefore, the average of both cycles gives a better performance and savings described in this paragraph.

#### 2.2.2. System with HP and Solar Collectors

In this case, there is the connection of a HP system with solar collectors: the idea is to assist HP with solar gains and to increase the performance efficiency, where the system is improved with PCM and optimal system controls. In this respect, Youssef et al. [52] designed a test rig with an indirect solar assisted HP (IDX-SAHP) system with three loops: solar thermal, solar assisted HP and load profile. The PCM tank was installed in the solar thermal loop system to store the excess of solar energy. Stored heat then served as a heat source for the HP, when required (night-time or low solar irradiance). A PCM-based heat exchanger was used as an LTS storage tank with a total volume of 340 L (300 L water and 40 L PCM), where the PCM (paraffin with melting temperature at 17 °C) presented a volume of 12%. Comprehensive measurements were carried out in different weather conditions. The results showed that the system could efficiently meet the daily DHW demand and the PCM storage integration exerted a significant effect on the system stability and performance efficiency. Thus, compared with systems without PCM tank integration, the average COP of the IDX-SAHP system with a PCM tank increased by 6.1% (COP = 4.99) and 14.0% (COP = 4.8) on sunny and cloudy days, respectively.

#### 2.2.3. System with HP and PVT Panels

Besagni et al. [53] have similarly analyzed solar-assisted HP in a research similar to that described in Section 2.2.2. The difference was that they coupled hybrid photovoltaic/thermal (PVT) panels with a multifunctional and reversible HP. The system was installed in a detached house in Milan and it was equipped with “air-source” and “water-source” evaporators, connected in series. In addition, the PVT panels were used, by employing two storage tanks, which were not PCM-, but water-based. One storage tank (186 L) was for DHW and the other intermediate storage (300 L) was to provide a heat source to the “water-source” evaporator. The system gave promising results, as the COP increased by 34% with the use of intermediate storage as the “water-source” evaporator. Further, approximately 63% of the heat needed for the DHW tank was provided by the PVT panels. Therefore, in comparison with a regular air-to-water HP, this system had 15.4% lower daily-averaged electricity consumption. According to a previous research, it would be intriguing for this system to test the integration of PCM in the tanks and observe the improvements, compared with the water-only store.

#### 2.2.4. System with Solar Collectors

The intermittence of solar radiation creates a gap between energy demand and supply, which requires the use of efficient TS to make solar energy more energetically justified [54]. This makes it one of the most analyzed technologies, where examples of integrated PCMs in different parts of storage tanks are presented below.

The first example is a solar domestic hot water tank with a water volume of 148 L and 23 L of PCM in the mantle between the outer wall of the tank and the insulation layer. The tank was designed to provide DHW for residential dwellings, through a combination of solar and auxiliary heating, concurrently using PCM on the basis of cheap sodium acetate trihydrate (SAT) as a thermal battery to shave off peak auxiliary power or to work under power outage [55]. The authors evaluated the influence of PCM with a melting point at 58 °C on heat content and heat charge and discharge in several tests. They found out that 13.5% of PCM in a mantle of the tank increased the heat release for the DHW supply by 32%, compared with the water-only store. This is valid for a temperature difference between 87 °C when the TS is fully charged and a temperature of 40 °C, since fresh water is supplied to the TS at a temperature of 17 °C. Discharge efficiency was 70%, which means that the PCM was utilized up to 70%. A study also indicated that the PCM composite was stable without performance degradation after 16 test cycles lasting for over three months. This indication is also an important parameter when designing a TS solution with PCM. Contrary to the presented study, Lu et al. [56] integrated two different PCMs in embedded containers inside a 140 L water storage tank for DHW. They used a maximum of 22.8 kg of SAT with a higher melting point and 15.6 kg of lauric acid with a lower melting point. In order to fully use latent heat, they embedded the PCM inside the tank and placed the PCM with a higher melting point (58 °C) in the upper half of the tank and the PCM panels with a lower melting point (44 °C) in the lower half. For the operating temperature of the TS, which was between 70 °C and 40 °C (30 K) with fresh water supply to TS at 23 °C, the experimental investigation revealed that 19% of the PCM in the tank increased the heat release by 39%, compared with the water-only store. The authors also made an economic analysis, where they calculated a return period of five years, which makes this prototype very intriguing for application in a real environment.

Fazilati et al. [57] also introduced PCM into a solar water tank for DHW, but they used paraffin wax embedded in 180 spherical capsules of 38 mm in diameter. A jacketed shell heat exchanger with a volume of 9.5 L was used, and the PCM capsules were integrated inside. The paraffin had a temperature melting point of 55 °C. The capsules represented 54% of the volume, however the amount of the PCM in the tank was not provided and can only be assumed. To enhance the low thermal conductivity of paraffin, a 380 mm long copper wire with a 0.3 mm diameter was inserted in each capsule. The aim of their study was to compare energy and exergy efficiency with or without PCM at different temperatures of the charged TS. The temperatures of the charged TS were 40 °C, 60 °C and 80 °C, with an inlet temperature of around 30 °C in each test. As most of the other studies, the research considered also confirmed that the inclusion of PCM has a positive effect in comparison with the water-only store, since the heat storage capacity was increased by up to 39% and the exergy efficiency by up to 16%. The supply of hot water with a specified temperature was 25% longer compared with the water-only store. However, the performance of the water tank with the PCM tested at 40 °C was not greater than the water-only store because the paraffin with a melting point of 55 °C did not melt. Therefore, a lower specific heat of paraffin in sensible heat in comparison with water led to a decreased performance. A similar study was done by Raul et al. [58], who also analyzed spherical capsules in a solar water tank. The organic material A164 with a melting point at 169 °C was used in 209 capsules that were integrated in a water tank with a volume of 8.2 L, where PCM represented 31% of the volume. Both the mathematical model and the prototype were made, and the main conclusion of the parametric study was that with a smaller diameter of the capsules and a higher porosity, the melting and solidification processes are faster; however, a higher pressure drop occurs, which could lead to significant consumption of the circulation pump. In addition, it has been found that the inlet temperature of water in the tank, when discharging heat from the LTS, has a bigger influence on the overall performance of the LTS than when charging the LTS. Charge efficiency increased up to 66% in the case of increased inlet water temperature vs. charging the LTS. However, in the case of decreasing the inlet temperature when discharging heat from the LTS, efficiency increases up to 77% (the operating temperature of the LTS was between 185 °C and 120 °C). The parametrical analysis also showed that an increase in flow rate of 2.2 L/min produces a marginal improvement in discharge efficiency.

Contrary to the works presented in this sub-section, where the authors mostly focused on temperature responses and the increase in energy density in order to improve supply time, Englmair et al. [59] mainly focused on reducing the energy consumption in buildings. They proposed a solar combi-system with short- and long-term heat storage with 22.4 m^2^ evacuated tubular collectors, a 735 L water tank and four compact PCM units, each containing 150 L of SAT. The idea was to cover the heat demand of a Danish single-family passive house. With the application considered, 69% of the annual solar fraction for heat supply was achieved. From April to September, 100% of the solar fraction was achieved for the DHW supply. A more detailed review on cascaded PCMs in solar water collector storage tanks can be found in the work by Zayed et al. [60].

#### 2.2.5. System with Electrical Heating

Finally, PCMs can also be integrated in electrical hot water cylinders [61], so that low-cost electricity during low-peak periods can be used. The authors experimentally tested encapsulated PCM (salt hydrate TH58 with a melting point at 58 °C) placed in 57 vertical tubes for validation purposes, and then simulated different cases where the diameter and thickness of the pipes were changed. Naturally, this depends on the target according to which the system will be optimized for, but 22% of PCMs in a 180 L tank at a water temperature of 70 °C and discharged at 15 °C increased the hot water release by 20%, compared with the water-only store. Consequently, the coverage of DHW demand increased from 40% to 55%.

An overview of the heating technologies with PCM applications that are described in Section 2.2. is presented in Table 2. In this review, a smaller amount of PCM was implemented in TS solutions, compared with cooling technologies. The maximum percentage of PCM volume in TS among all applications was 45% (the eighth example), but from an energy density point of view, this case was one with the lowest energy density. The reason could be that the LTS was not completely charged, since it was heated at 58 °C, whereas the melting point of PCM in the system was also 58 °C. In addition, the temperature difference of the TS was much smaller than in other cases, but compared with the third example, where the water-only store had a lower temperature difference but a higher energy density, this is not the reason for the low amount of energy stored. Nevertheless, all the applications provided an improvement with the inclusion of the PCM in the TS, and in this case, the majority of examples compared the inclusion of PCMs in a TS with a water-only store, which provided an appropriate comparison.

The system which stands out from review in this sub-section (Table 2 is presented in the sixth example, with two PCMs embedded in a water storage tank. In this TS, only 19% of PCM at 30 K of the TS temperature difference increased the heat release by 39%. Another very intriguing application is in the fourth example, where the PCM in a mantle (between the outside wall of the tank and the insulation layer), which represented 13.5% of the TS volume, increased the heat release by 32% at 47 K of the TS temperature difference. This TS application also had the highest utilized energy density among all TS applications in heating technologies. The sixth and ninth examples provided a 25% and 20% increase in heat release with the inclusion of 30% and 22% of PCM, respectively. These improvements show a lot of potential, especially if released heat is taken into account and is compared with cooling technologies, which have a higher percentage of PCM, but a lower heat release. The main reason lies in the higher temperature difference, which contributes to the higher heat transfer rate between PCM and water. Further, a smaller amount of PCM could be the reason for the complete phase-change process—and therefore the higher heat release—because of the higher effective heat transfer surface.

### 2.3. Space Cooling and Heating

Systems providing both cooling and heating (Table 3) are primarily solar-based, and the interest in solar thermal cooling has been recently growing for HVAC applications [62,63,64]. The most common is sorption cooling, and one of the examples is the study done by Behi et al. [65], who carried out a numerical and experimental evaluation of a novel solar-collector-integrated sorption module for the cogeneration of cold and heat. They analyzed the sorption module with LiCl-H_2_O and a built-in TS system for cooling and heating. The feasibility of the use of PCM for cold and heat storage, which was coupled with a condenser/evaporator, was investigated. Paraffin was used as the PCM, with a melting point at 27 °C for heat storage (RT27) and with a melting point at 11 °C for cold storage (RT11). The results showed that the cooling output of a 100 L cold storage tank between 9 and 20 °C was 331 Wh, and the heat output of a 100 L heat storage tank between 27 and 20 °C was 425 Wh. Overall, in the course of the operational period, the average COPs of the system were 0.36 for cooling and 0.42 for heating. The analyzed integration of PCM with the sorption module upgrades the supply system, providing cooling/heating with a higher flexibility. However, there is no information of the amount of PCM in the tank. Considering the operating temperature level, the melting point of both PCMs and the low amount of heat and cold stored, it could be concluded that PCM did not completely undergo a phase-change process, or that there is a very low percentage of PCM.

TS can also be integrated into systems with solar-driven adsorption chillers and radiation heating. One example is the research by Mendecka et al. [66], where the authors thermodynamically analyzed the overall performance of the system without TS, with water storage and with PCM storage (PCM-based shell and tube heat exchanger). The system was designed to produce chilled and hot water for cooling and heating purposes for single-family buildings, utilizing solar energy and/or natural gas. The mass and energy balances were resolved in TRNSYS and Matlab. The results showed that the introduction of a PCM shell and tube storage (500 L) increased the utilization of solar energy by 7%, and increased annual efficiency by approximately 5% compared with a water-only store. The results show that the PCM did not provide a significant improvement; however, it has to be stressed that the water storage tank had twice the volume of the PCM storage tank. An overview of heating and cooling technologies with the PCMs used as TS is summarized in Table 3.

From the review presented throughout Section 2, it can be seen that the inclusion of PCMs improved the performance of every system. However, not every research in the review used an appropriate comparison with the baseline system, whereas a comparison with an alternative solution—such as a water storage tank—would have produced a more credible outcome. Furthermore, the discharge efficiency of LTS proved to be lower than the discharge efficiency of the water TS because of the heat transfer resistance between the PCM and the heat transfer fluid. This can be solved with a higher heat transfer surface, or a higher temperature difference (as was the case in the heating technologies) of the LTS when discharging heat or cold. Therefore, it was found that a decreasing inlet temperature of the LTS tank has a higher influence on the LTS efficiency when discharging heat than increasing temperature when charging the heat in the LTS. In addition, a lower percentage of PCM provided a greater utilization of the PCM because of a higher discharge efficiency. That means that a lower percentage of PCM probably had a greater effective heat transfer surface, which improved the heat transfer rate between the PCM and the water. However, this does not confirm that the integration of only a small amount of PCM should take place, but rather that the range of the operating temperature which was proven to also be effective should be considered. It might be true that PCM improves the energy performance of systems, but an economic analysis should also be considered when deciding to implement PCMs in a real environment. In the reviewed technologies, only one economic analysis was made, which stated a payback period of five years. The economic analysis is most definitely something that is missing in the investigation of the case studies with PCM application in order to obtain the feasibility of implementation in the real environment. In addition, the cycling stability of PCM was not mentioned in most of the research, since only in one study a 16-cycle test was performed, which does not confirm a long-term cycling stability. With regard to the types of PCM, it is hard to define which performed better because in some cases the same type of PCM performed better in one application than in another. In order to clarify this issue, the investigation of different types of PCM should be performed within the same application, and also cycling stability and economic analysis should be taken into account. Material and thermal properties of PCM have not been listed for both the liquid and solid state of every case, due to the lack of information of reviewed articles of considered cases. The material and thermal properties of PCMs considered for applications in heating and cooling systems can be found in the literature that review PCM technology such as Mehling et al. [67] and Paksoy et al. [68]. Under Task 32 of the Solar Heating and Cooling Programme at the International Energy Agency, an inventory of PCMs has been made [69] with an overview of the material and thermal properties of PCMs. In the work of Cabeza et al. [70], a large amount of PCMs with material and thermal properties are listed.

According to the present review, it can be seen that any kind of TS integration in the system improves the performance of the heating or cooling technologies, and consequently the system itself. Therefore, the TS can improve the COP/EER of HPs with load shifting by operation under lower electricity fares and favorable external conditions. It can also improve the COP/EER by providing favorable temperature conditions as a heat source for HPs. The latter is done by utilizing solar energy to increase the temperature of the heat source for heating purposes, or by utilizing night-time operations to decrease the temperature of the heat sink for cooling purposes. In addition, it improves the utilization of the solar fraction, which makes the TS solution an indispensable part of the system utilizing RES.

### 2.4. Indirect Electrical Storage for Space Cooling and Heating

In this section, a practical novel application of LTS coupled with a HP system for indirect electrical storage in the form of heat and cold is reviewed. The application of the inclusion of PCM in the system is analyzed in more detail because of its potential for a more sustainable exploitation of solar energy through the energy storage of excess electricity also in the form of heat or cold.

More in detail, the HEART (Holistic Energy and Architectural Retrofit Toolkit) project, funded in the context of H2020-EU.2.1.5.2 [14], introduces a technological advancement beyond the current state-of-the-art, by proposing system innovations used in a holistic approach to the energy retrofit of residential buildings. In particular, it aims to maximize the self-consumption of RES in existing residential buildings, by coupling a PV system with DC-HPs that are connected with a TS system (Figure 2). The latter is thus able to store the excess of PV energy as a thermal form of energy at a low temperature, which can be exploited later, as needed. This allows a reduction in the mismatch between solar radiation and thermal loads. It is critical to emphasize that the heat or cold produced in a centralized DC-HP is stored at low temperature in the TS unit, since each apartment contains a small-size fan coil with a DC compressor (local HP), used to increase the thermal power (coming from the centralized HP or LTS), according to the thermal demand of each room. This allows to minimize heat losses in the existing distributions pipes, in order to avoid condensation in the cooling mode and to increase the COP/EER of the centralized HP.

One of the investigations that took place within the HEART project was the enhancement of the energy density of the TS, where—as described in the previous paragraph—the TS plays an important role for storing heat or cold over time when solar energy is not available. Therefore, an experimental investigation of encapsulated PCM implemented in the water storage was made. In the experimental prototype, 43 L of paraffin was encapsulated in cylindrical modules, which were implemented at the top of a 280 L water storage tank. Therefore, PCM represented 15% of the volume in the tank. The paraffin used in the system was RT28HC and had a phase-change temperature between 27 and 29 °C, with the main peak at 28 °C. The storage tank was heated at 32 °C and it was discharged with inlet water at 21 °C. The discharge of the storage tank was stopped when it reached the temperature of 25 °C, where the temperature difference between the start and end temperatures was 7 K. The results of the tests show that the LTS unit increased the desired level of water temperature by 79% compared with the use of a water-only store. Thus, at a smaller temperature difference, the integration of PCM proved to have a positive effect because of its higher energy density, which can be stored during the phase-change [71,72].

## 3. Maturity of PCM for Thermal Storage

Since a major barrier to the development of TS technologies for the use of RES is cost uncertainty [73], the economic evaluation plays a vital role in the deployment of TS technologies. That is why this section contains a review of the readiness of PCMs. The objectives from roadmaps are also reviewed, in order to obtain information for the further development of PCMs and the future price, which is crucial for PCMs to be cost-effective. The latter is also important because the reviewed literature (with the exception of one) from previous sub-sections did not include the economic analysis of the application with PCM. Since the reviewed applications with PCMs showed an improvement, it is important to review the market maturity and the possibilities of a cost reduction.

According to the published roadmap of the European Technology Platform on Renewable Heating and Cooling [74], the identification and quantification of the key performance indicators (KPIs) for TS in the heating and cooling of buildings have been defined. For water storage tanks, a cost reduction for a 1000 L tank is defined in 2020 from EUR 400–900 to EUR 300–400, while for the LTS application, the goal is to develop salt hydrate PCMs costing less than 2 EUR/kg, again in 2020. In addition, the goal is to develop new heat exchangers with integrated PCMs, to market several applications in 2020 and to develop sensors to determine the PCM state of charge.

The technology maturity curve in Figure 3 consists of the technology readiness level (TRL) and the stage of market development of energy storage technology for small-scale energy systems. Nguyen et al. [75] obtained TRL results by examining different reviews of energy storage technologies and marked the current state of technology development. Furthermore, the stages of market development are evaluated according to research and development, demonstration, growth and maturity. Figure 3 shows that in TS technology, STS is currently the most mature technology for small-scale energy systems. Hybrid systems with macro-encapsulated PCMs integrated in a water TS tank for increasing energy density are in the demonstration stage, and some of them have already been introduced on the market. PCMs are mostly chosen based on cost, but further advancements are possible in this area, according to several research projects [74]. According to the Roadmap towards 2030 of the European Energy Storage Technology Development [76], a further research and development of PCMs should focus on new materials for the LTS, on the improvement of the thermo-physical properties of PCMs, on the identification of advanced heat transfer mechanisms for charging and discharging and on the development of advanced HTFs for thermal electricity storage systems combining heat transfer and heat storage. Furthermore, EASE states that the target for the technology is to have in 2030 a specific investment cost below 50 EUR/kWh for compact latent heat and thermo-chemical storage.

## 4. Conclusions

The review of LTS applications in hydronic systems for space heating and cooling shows that there is ongoing research on PCM applications, from the systemic aspects to the enhancement of the HP performance or the use of the solar energy potential. With the integration of PCMs, the performance of almost every system under review has improved. It has been found that systems with a HP (either for space heating or space cooling) which are equipped with LTS have on average an approximately 14% higher heat release in comparison with a water-only store. In systems with solar collectors using LTS, the heat released from LTS increased up to 39% compared with water storage tanks. In solar-driven sorption cooling and heating systems, the integration of PCMs to the sorption module upgraded the supply system, providing cooling and heating with a higher flexibility, but there was no comparison with a baseline system, and also the energy density was really low, which could be due to the temperature level that was mismatched with the melting point of the PCMs. However, in systems with solar-driven adsorption chillers and radiation heating, the integration of a PCM-based shell and tube storage tank showed promising results. In comparison with a water-only store, the inclusion of PCMs increased the solar fraction by 7% and annual efficiency by approximately 5%. In addition, it should be noted that the water storage volume was twice the size of the LTS.

Within the HEART project, the inclusion of 15% PCM at a TS temperature difference of 7 K, increased heat supply by 79% compared with a water-only store. Thus, at smaller temperature differences, the integration of PCMs proved to have a positive effect because of a higher energy density that can be stored during the phase-change. However, it is to be noted that, from the review of other technologies in this article, the criticality might appear exactly at small temperature differences, where the PCM did not completely undergo a phase-change. At this point, it goes without saying that a high heat transfer rate from PCM to HTF needs to be provided. Furthermore, it should be pointed out that the problems of PCMs have not been discussed in the reviewed literature. More or less, only comparisons to the base line system or only evaluations of performance were carried out. There was no information on the achieved potential of PCM applications, nor was the state-of-charge during the process cycle of releasing or accumulating the heat defined. Therefore, further research on the numerical investigation of LTS and its validation within a real environment operation would be useful in order to obtain information on the state-of-charge of the LTS which is needed for an optimum performance and to achieve the full potential of PCMs.

Besides the current state of research and the roadmaps on LTSs, the decrease in the price of PCMs and a few already available LTS solutions on the market indicate that LTS solutions for the heating and cooling of buildings are progressing from the demonstration stage towards mature technology. However, the current cost of such technology restricts its wide diffusion on the market.

All in all, it can be concluded that the indirect storage of excess PV energy through a HP in TS has not been much of a subject in recent research. In particular, the integration of PCMs has not been considered, despite the fact that they have proven to be promising from an energy performance perspective. This opens the field of research and possible improvements in higher fractions of RES self-consumption in residential buildings, such as the solution proposed in the HEART project. This project showed that renewable energy, in the form of heat or cold, can be stored in PCMs in a smaller volume compared with a water-only store. With this, the problem of space limitation in the existing technical rooms of residential buildings can be solved.

## Figures and Tables

**Figure 1 materials-13-02971-f001:**
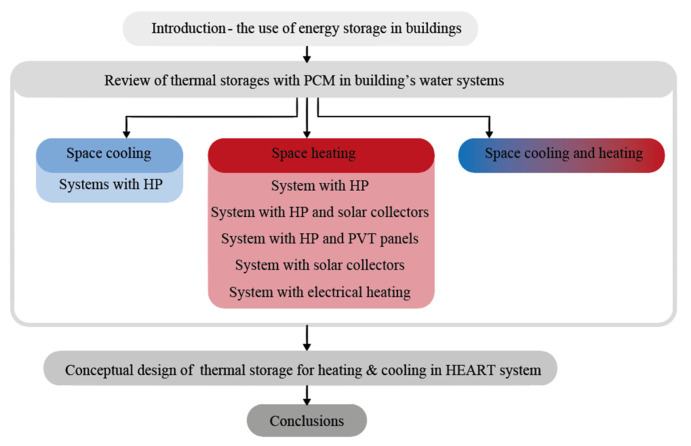
Particular sections of the review.

**Figure 2 materials-13-02971-f002:**
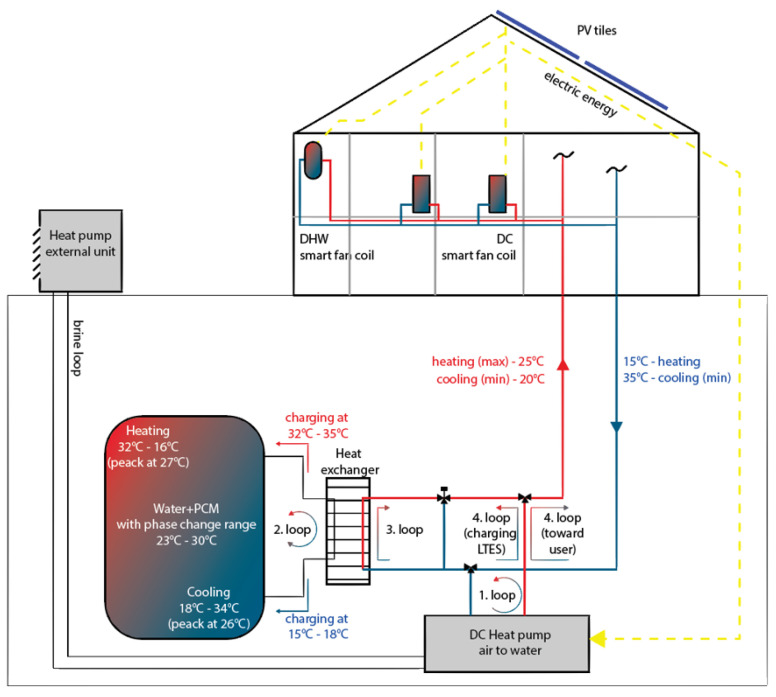
Scheme of the HEART system. 1. loop—charging of the latent thermal storage (LTS) unit with a heat pump (HP); 2. loop—circulation of water through the LTS unit during the charging or discharging process; 3. loop—discharging of heat/cold from the LTS unit into the heating/cooling system; 4. loop—the HP is simultaneously charging the LTS unit and the heating/cooling system.

**Figure 3 materials-13-02971-f003:**
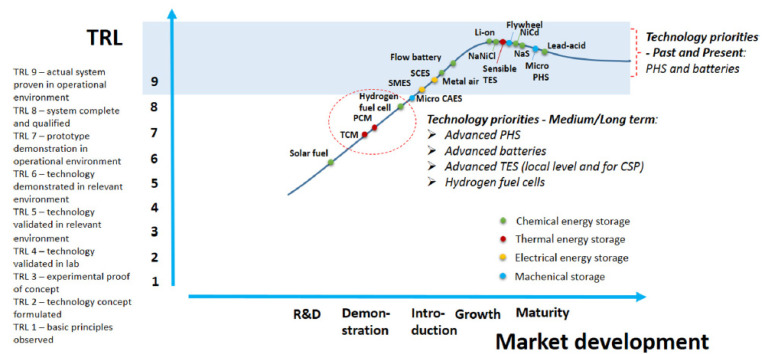
Maturity of energy storage technologies [75].

**Table 1 materials-13-02971-t001:** Overview of cooling technologies with phase-change materials (PCMs).

Cooling	Technology	Encapsulation/PCM Type	Material Properties	Accumulation	Size	Energy Density	SPF
PCM thermal energy storage tanks in heat pump system for space cooling [46]	Water/water heat pump	Plates encapsulation Salt hydrate (S10) Melting point: 10 °C	h = 155 kJ/kg (228 kJ/L) c_p_ = 1.9 kJ/kgK λ = 0.43 W/mK ρ = 1470 kg/m^3^	Stored: 2.79 kWh (water) 3.88 kWh (LTS) Released: 2.32 (water) 2.61 kWh (LTS)	56 L water + 48 L PCM (46%)	25 kWh/m^3^ (PCM) 22 kWh/m^3^ (water)	/
Improvement of a heat pump based HVAC system with PCM thermal storage for cold accumulation and heat dissipation [47]	Water/water heat pump	FlatICE panels Salt hydrate (S10) Melting point: 10 °C (cold) and 27 °C (heat)	h = 183 kJ/kg (280 kJ/L) c_p_ = 2.2 kJ/kgK λ = 0.54 W/mK ρ = 1530 kg/m^3^	/	/	/	/
Free cooling potential of a PCM-based heat exchanger coupled with a novel HVAC system for simultaneous heating and cooling of buildings [48]	Reversible air-to-water heat pump with dry cooler	PCM-based heat exchanger Plant-based PCM Melting point: 18 °C	h = 192 kJ/kg c_p_ = 1.47–1.74 kJ/kgK λ = 0.15–0.25 W/mK ρ = 860–950 kg/m^3^	1.1 MWh	21 m^3^	52 kWh/m^3^	SPF = 5.6 (HP + LTS)
Geocooling with integrated PCM thermal energy storage in a commercial building [49]	Geocooling	Spherical capsules Mix of water & nucleating agents Melting point: 0 °C	h = 333 kJ/kg c_p_ = 1.47–1.74 kJ/kgK λ = 0.56–2.2 W/mK ρ = 1000–917 kg/m^3^	Stored: 27 kWh (one unit)	200 L water + 300 L PCM (60%) -one unit	54 kWh/m^3^	SPF1 = 3.0 (base) SPF2 = 5.0 (no PCM) SPF3 = 5.1 (with PCM) 41% and 2% increase
Energy saving performance assessment and lessons learned from the operation of an active PCMs system in a multi-story building in Melbourne [50]	Adiabatic cooler	5120 FlatICE PCM panels Salt hydrate Melting point: 15 °C	h = 160 kJ/kg c_p_ = 1.9 kJ/kgK λ = 0.43 W/mK ρ = 1510 kg/m^3^	Total: 1500 kWh (max. in winter) Latent: 1307 kWh (theoretically)	20 m^3^ water + 20 m^3^ PCM (50%)	36 kWh/m^3^	/

**Table 2 materials-13-02971-t002:** Overview of heating technologies with PCMs.

Heating	Technology	Encapsulation/PCM Type	Material Properties	Accumulation	Size	Energy Density	COP
Experimental research of an air-source heat pump water heater using water-PCM for heat storage [51]	Air-source HPWH	Around condenser coil Paraffin (RT44HC) Melting point: 43 °C	h = 255 kJ/kg λ = 0.2 W/mK ρ = 760–860 kg/m^3^	Water: 6.4 kWh LTS: 7.3 kWh (14% increase)	138 L water + 11 L PCM (7.5%)	49 kWh/m^3^ ΔT = 40 K	3.74 (HP) 5% increase
Effects of latent heat storage and controls on stability and performance of a solar assisted heat pump system for domestic hot water production [52]	Water-source HP (DHW)	PCM heat exchanger tank Paraffin Melting point: 17 °C	h = 260 kJ/kg λ = 0.2 W/mK c_p_ = 2 kJ/kgK ρ = 770–880 kg/m^3^	77.90 kWh (test day)	300 L water + 40 L PCM (12%)	/	System COP: 4.99 (sunny) 4.8 (cloudy) 6%–14% increase
Field study of a novel solar-assisted dual-source multifunctional heat pump [53]	Air-source HP (DHW)	Water DHW: 48–58 °C Tank: 22–38 °C	c_p_ = 4.18 kJ/kgK λ = 0.64 W/mK ρ(53 °C) = 987 kg/m^3^ ρ(30 °C) = 995 kg/m^3^	DHW: 4 kWh	186 L (DHW) + 300 L (tank)	DHW: 21.5 kWh/m^3^	Monthly: 3.75 (max) 2.47 (min) 34% increase
Thermal performance assessment and improvement of a solar domestic hot water tank with PCM in the mantle [55]	Solar water tank—DHW	PCM in a mantle, Sodium Acetate Trihydrate (SAT) Melting point: 58 °C	h = 262 kJ/kg λ = 0.54 W/mK c_p_ = 3.22 kJ/kgK ρ = 1450 kg/m^3^	Total: 12 kWh PCM: 3.8 kWh	148 L water + 23 L PCM (13.5%)	Water:12 kWh/m^3^ LTS: 92 kWh/m^3^ T = 87–40 °C	/
Study on the performance of heat storage and heat release of water storage tank with PCMs [56]	Solar water tank—DHW	Embedded containers SAT (58 °C), Lauric acid (44 °C)	h = 262 & 180 kJ/kg λ = 0.54 W/mK c_p_ = 3.22 & 2.5 kJ/kgK ρ = 1450 & 880 kg/m^3^	Water: 5.9 kWh LTS: 8.3 kWh 39.2% increase	110 L water + 30 L PCM (19%)	57 kWh/m^3^ T = 70–40 °C	/
Phase-change material for enhancing solar water heater, an experimental approach [57]	Solar water tank—DHW	180 spherical capsules Paraffin Melting point: 55 °C	h = 187 kJ/kg λ = 0.2 W/mK c_p_ = 2–2.15 kJ/kgK ρ = 790–910 kg/m^3^	25% increase	Approx.: 6.7 L water + 2.8 L PCM (30–40%)	/	/
Modeling and experimental study of latent heat thermal energy storage with encapsulated PCMs for solar thermal applications [58]	Solar water tank	209 spherical capsules Organic material A164 Melting point: 169 °C	h = 250 kJ/kg λ = 0.45 W/mK c_p_ = 2.01 kJ/kgK ρ = 1500 kg/m^3^	0.5 kWh	5.7 L water + 2.5 L PCM (31%)	61 kWh/m^3^ T = 185–120 °C	/
A solar combi-system utilizing stable supercooling of sodium acetate trihydrate for heat storage: Numerical performance investigation [59]	Solar water tank—DHW	Compact storage tank Sodium Acetate Trihydrate Melting point: 58 °C	h = 180–200 kJ/kg λ = 2–5 W/mK c_p_ = 2.5 kJ/kgK ρ = 1350–1400 kg/m^3^	4 × 2.6 kWh = 10.4 kWh	735 L water + 4 × 150 L PCM (82%)	17.3 kWh/m^3^ T = 58–45 °C	/
Thermal analysis of including phase-change material in a domestic hot water cylinder [61]	Electrical hot water cylinder	57 PVC tubes Salt hydrate TH58 Melting point: 58 °C	h = 185 kJ/kg λ = 0.54–1.09 W/mK c_p_ = 2.88–4.19 kJ/kgK ρ = 1290–1400 kg/m^3^	Water-only: 11.9 kWh LTS: 14.3 kWh (20% increase)	141 L water + 39 L PCM (22%)	79.6 kWh/m^3^ T = 70–15 °C	/

**Table 3 materials-13-02971-t003:** Overview of heating and cooling technologies with PCMs.

Heating and Cooling	Technology	Encapsulation/PCM	Material Properties	Accumulation	Size	Energy Density	COP
Evaluation of a novel solar driven sorption cooling/heating system integrated with PCM storage compartment [65]	Sorption cooling LiCl-H_2_O	Paraffin: RT27 (heat), RT11(cold) Melting point:27 °C (heat), 11 °C (cold)	h = 149 (RT27) & 160 (RT11) kJ/kg λ = 0.2 W/mK c_p_= 2 kJ/kgK ρ = 760–800 kg/m^3^ (RT27) ρ = 770–880 kg/m^3^ (RT11)	331 Wh (cooling) 425 Wh (heating) 756 Wh (overall)	2 × 0.1 m^3^	Cold: 3.3 kWh/m^3^ (ΔT = 20–9 °C) Heat: 4.3 kWh/m^3^ (T = 27–20 °C)	Cooling: 0.36 Heating: 0.42
Energetic and exergetic performance evaluation of a solar cooling and heating system assisted with thermal storage [66]	Adsorption chiller + radiation heating	Shell and tube heat exchanger Paraffin Melting point: 62 °C	h = 200 kJ/kg λ = 0.162–0.35 W/mK c_p_ = 2.05–2.26 kJ/kgK ρ = 825–880 kg/m^3^	36 kWh	LTS: 0.5 m^3^ Water: 1.1 m^3^	72 kWh/m^3^ (T = 80–55 °C)	Annual system en. efficiency: No TS: 24.2%, Water: 31.9%, LTS: 33.4%

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
