# Peer review of "Phase-Change Materials in Hydronic Heating and Cooling Systems: A Literature Review"

_materials, 2020, doi:10.3390/ma13132971_

Round 1

Reviewer 1 Report

The submitted paper is focused on the study of using the phase change materials in hydronic heating and cooling systems. The use of PCM materials for passive and active heating or cooling of buildings represents a modern and promising method, how to significantly reduce the economic costs and environmental burden associated with maintaining the required air temperature inside buildings.Paper is well written. The authors present an extensive interesting review with links to many scientific papers dealing with the studied issues. The results are described in detail and presented by the relevant tables. Therefore I recommend accepting the paper for publishing.

Other comments:
Page 1, Nomenclature: There should be Kelvin as a unit of a temperature difference.

Author Response

Dear reviewer,

we appreciate your careful reading of our manuscript and your comments.

Attached please find the document where we address your comments.

Thank you very much for your attention to our manuscript.

Sincerely yours.

Reviewer 2 Report

The paper is entitled "Phase change materials in hydronic heating and colling systems: literaure review and proposal for a novel application" but in this paper there are not discussed the problems of phase change materials.

The paper contains mainly descriptions of the different heating and cooling systems. Besides it should contain more schemes and illustrations. It is for me strange that authors do not give in the Acknowledgements the title of the grant and only inform that the project was realized within the  framework of Horizon 2020 Programme through the numer 768921-HEART.

Summarizing the paper describes the systems of different methods of heating and cooling and generally it does not suit for the publication in the Journal "Materials". The authors should look for the another Journal dealing with the heating and cooling processes.

Author Response

(The authors gave the same response as above.)

Reviewer 3 Report

The present manuscript focuses on a review of publications investigating hydronic heating and cooling  systems using PCMs. Additionally it dedicates a small section in a discussion concerning the TRL of latent heat storage and an additional section to introduce the conceptual design of latent thermal storage for heating & cooling. The review alone is an interesting piece of work that is certainly useful for the community. Subsection 2.4 and section 3 seem a bit  "forced-in" and unconnected to the review. My main comment is therefore that I would suggest that the authors either present the connection between these different parts more strongly and clearly or they remove subsection 2.4 and section 3 altogether and focus on the review part.

Some minor comments can be also found below:

  • Many small typos and grammar mistakes.
  • Page 3 line 105:In my opinion stating that LTS has advantages over water storage in terms of "heat transfer efficiency" is quite misleading. If the authors want to include the comparison to a water storage they should mention also the disadvantages of LTS.
  • Page 4 line 132: normally tables are introduced before they are presented.  I would suggest to introduce the tables later in the manuscript. For example Table 2 is first mentioned in page 4 but it is actually presented in page 11.

Author Response

(The authors gave the same response as above.)

Reviewer 4 Report

Review Report on Manuscript ID: materials-820924

Title:

Phase change materials in hydronic heating and cooling systems: literature review and proposal for a novel application

Authors:

Rok Koželj*, Eneja Osterman, Fabrizio Leonforte, Claudio Del Pero, Alessandro Miglioli2, Eva Zavrl, Rok Stropnik, Niccolo Aste, Uroš Striti

General statement

This is an interesting review paper concerning latent thermal storages in hydronic systems for heating, cooling and domestic hot water in buildings (concept of nearly zero energy buildings). Various cogeneration systems were discussed in the paper among them systems with heat pumps, systems with heat pumps and with/without solar collectors, systems with heat pumps and photovoltaic panels, systems with electrical heating. The smart energy systems discussed in this work are subject of HEART project (HORIZON 2020) which according to goals of EU, consumption of primary recourses should be reduced by 80% by the end of 2050 year. The work is written carefully and comprehensibly. The literature cited contains 77 items, of which 3 are the authors' own works. In my opinion, the work fully meets the requirements of a good review article (i.e. synthetic approach to the problem, sufficient level of detail, state of the art of cogeneration systems, percentage energy savings, future solutions)

Title – informative and catching.

Abstract – informative and straight to the point.

Introduction – informative, based on relevant and up to date literature.

Main body – well-chosen text, figures readable of good quality, tables contain relevant and sufficient information.

Conclusions – adequate.

References – relevant and up to date.

Author Response

(The authors gave the same response as above.)

Reviewer 5 Report

The paper analyzes Phase change materials in hydronic heating and cooling systems.

From the analysis of the information presented in the article, I found the following:

- the keywords should be filled in because they do not cover the entire topic presented in the paper;

- the introduction is quite long, it should be more withdrawn, especially since section 2 of the paper presents almost the same information as the introduction;

- the research methodology is unclear and needs to be substantially improved, especially if we take into account the fact that a large part of the paper is review type. It must be presented the way in which the selection of the analyzed works was made (search by certain keywords, etc.);

- Tables 2 and 3 must be completed with other information presented in other works in the field;

- greater emphasis must be placed on the proposed new application (more detailed presentation, advantages, novelty in relation to the information currently known);

- future research directions are not presented.

Thus, the article cannot be accepted for publication in this form. Substantial changes to the article are required so that it can be published.

Author Response

(The authors gave the same response as above.)

Round 2

Reviewer 5 Report

The authors responded to the made comments in review. The article can be published in the presented form.

Author Response

Dear reviewer,

Thank you very much for your attention to our manuscript.

We appreciate your time!

Sincerely yours.